# Transition-metal-free allylation of 2-azaallyls with allyl ethers through polar and radical mechanisms

Guogang Deng[1,3], Shengzu Duan[1,3], Jing Wang[1], Zhuo Chen[1], Tongqi Liu[1], Wen Chen[1], Hongbin Zhang[1✉], Xiaodong Yang [1✉] & Patrick J. Walsh [2✉]

Allylation of nucleophiles with highly reactive electrophiles like allyl halides can be conducted without metal catalysts. Less reactive electrophiles, such as allyl esters and carbonates, usually require a transition metal catalyst to facilitate the allylation. Herein, we report a unique transition-metal-free allylation strategy with allyl ether electrophiles. Reaction of a host of allyl ethers with 2-azaallyl anions delivers valuable homoallylic amine derivatives (up to 92%), which are significant in the pharmaceutical industry. Interestingly, no deprotonative isomerization or cyclization of the products were observed. The potential synthetic utility and ease of operation is demonstrated by a gram scale telescoped preparation of a homoallylic amine. In addition, mechanistic studies provide insight into these $C(sp^3)$–$C(sp^3)$ bond-forming reactions.

[1] Key Laboratory of Medicinal Chemistry for Natural Resource, Ministry of Education; Yunnan Provincial Center for Research & Development of Natural Products, School of Chemical Science and Technology, Yunnan University, Kunming, P. R. China. [2] Roy and Diana Vagelos Laboratories, Penn/Merck Laboratory for High-Throughput Experimentation, Department of Chemistry, University of Pennsylvania, Philadelphia, PA, USA. [3] These authors contributed equally: Guogang Deng, Shengzu Duan. ✉email: zhanghb@ynu.edu.cn; xdyang@ynu.edu.cn; pwalsh@sas.upenn.edu

Since the importance of the medicinal chemistry concept "escaping from the flatlands" gained appreciation, greater research efforts have been devoted to the formation of bonds between two C(sp$^3$) carbons[1,2]. The allylation of nucleophilic carbon centers is one of the most useful methods for the formation of C(sp$^3$)–C(sp$^3$) linkages[3–8]. As a result, it has been widely applied in the synthesis of bioactive compounds and natural products[9–15]. For decades, tremendous effort has been devoted to developing new and efficient methods for allylic alkylation. Most allylic alkylations fall into one of the two classes based on the nature of the allylic electrophile. When the electrophile possesses a potent leaving group, such as an allylic halide or pseudohalide, allylation reactions can be conducted in the absence of a catalyst. The drawback of these reactions, however, is the high reactivity of the electrophile, making selective reactions difficult. In cases where the allylic electrophile is less reactive, such as allylic acetates or carbonates, allylic alkylations can be performed with the assistance of a catalyst. This latter method has the advantage of more stable electrophilic substrates and has been widely employed with enantioenriched transition-metal catalysts[16–25], most notably in the Tsuji-Trost reaction[26–32]. The shortcoming of this approach is it generally relies on precious metal catalysts. Recent advances on radical allylation reactions have also been reported[33].

Allylic alkylation of carbanions can be used to prepare homoallylic amines. In particular Kauffmann and co-workers have prepared homoallylic amines by allylation of 2-azaallyl anions using allyl bromide as an electrophile[34]. Homoallylic amines are incredibly useful precursors for the synthesis of a vast number of biologically active molecules[11,35–48]. Economical methods to prepare homoallylic amines remain in demand. The use of allylic alkylations[49–56] or decarboxylative allylic alkylation reactions[56–61] to prepare homoallylic amines has been demonstrated with various transition-metal catalysts, including Ni, Pd, Cu, Zn, Ir, Rh, and Yb (Fig. 1a, b).

Our team[62–70] and other groups[71–84] have been interested in the functionalization of 2-azaallyl anions through an umpolung strategy. Recently, we discovered and developed a unique radical generation

approach[85] for the transition-metal-free C(sp$^3$)–C(sp$^2$) (Fig. 2a) and C(sp$^3$)–C(sp$^3$) bond formations enabled by 2-azaallyl species[86,87]. We found that deprotonation of N-benzyl imines 1 generated semi-stabilized 2-azaallyl anions that readily undergo single electron transfer (SET) with a variety of electrophiles[85], generating 2-azaallyl anion intermediates that are persistent radicals[88]. These species have now been isolated and characterized by electrochemical methods and X-ray crystallography[69]. Inspired by the work of Murphy and co-workers on organic super electron donors (SEDs)[89–96], we demonstrated that 2-azaallyl anions served as SEDs and enabled transition-metal-free C–C bond formation via reduction of aryl or alkyl iodides followed by radical recombination with the resulting 2-azaallyl radical (Fig. 2a). This SED approach was further used for the preparation of benzofurylethylamines (Fig. 2b) and isochromene derivatives via SET from the 2-azaallyl anion, radical cyclization, and finally intermolecular radical–radical coupling reactions[97,98]. Based on the unusual reactivity of 2-azaallyl anions, we were curious about their ability to react with allyl electrophiles that bear leaving groups that were generally categorized as poor in both organic chemistry and in the presence of transition-metal allylation catalysts[99–103].

Herein, we report a rare transition-metal-free C(sp$^3$)–C(sp$^3$) coupling of allyl phenyl ethers with 2-azaallyl anions (Fig. 2c). Specifically, we describe coupling of 2-azaallyl species with allyl phenyl ether electrophiles to furnish $S_N2$- and $S_N2'$-type allylation products in good yields. This allylation approach enables the synthesis of homoallylic amines bearing various functional groups (38 examples, up to 92% yield). It is noteworthy that the simple combination of base and solvent enable the transition-metal-free allylation to proceed efficiently. Furthermore, no deprotonation and isomerization or cyclization of products was detected. Mechanistic studies provide insight into these C(sp$^3$)–C(sp$^3$) coupling reactions and suggest that reactions can proceed by either polar or radical mechanisms, depending on the substitution pattern of the electrophile.

**a** Transition-metal catalyzed allylation of imines

R$^1$ = alkyl, aryl; R$^2$ = alkyl, aryl, Ts, Ns, $^t$BuS(=O)
X = Cl, Br, OCO$_2$Me, OPO(OEt)$_2$, OH, OBoc, BF$_3$K

**b** Pd-catalyzed decarboxylative allylic alkylations

Tunge's work

Chruma's work

**Fig. 1 General strategies for homoallylic amine synthesis. a** Transition-metal-catalyzed allylation of imines. **b** Pd-catalyzed decarboxylative allylic alkylations.

**a** Radical coupling with 2-azaallyl anions

**b** Radical cyclization-coupling reactions

**c** This work:

**Fig. 2 Transition-metal-free reactions of 2-azaallyl anions. a** SET with aryl or alkyl iodides followed by radical–radical coupling. **b** SET from 2-azaallyl anions, cyclization and radical–radical coupling to afford benzofurans. **c** Allylation of 2-azaallyl anions (this work).

**Table 1 Optimization of coupling of ketimine 1a and allyl phenyl ether 2a[a,b].**

| Entry | Base (equiv.) | Solvent | Conc. | Assay yield (%) |
|---|---|---|---|---|
| 1 | NaN(SiMe$_3$)$_2$ (3.0) | MTBE | 0.2 M | 64 |
| 2 | NaN(SiMe$_3$)$_2$ (3.0) | DME | 0.2 M | 10 |
| 3 | NaN(SiMe$_3$)$_2$ (3.0) | CPME | 0.2 M | 74 |
| 4 | NaN(SiMe$_3$)$_2$ (3.0) | THF | 0.2 M | 20 |
| 5 | NaN(SiMe$_3$)$_2$ (3.0) | Dioxane | 0.2 M | 0 |
| 6 | NaN(SiMe$_3$)$_2$ (3.0) | DMSO | 0.2 M | 0 |
| 7 | NaN(SiMe$_3$)$_2$ (3.0) | DMF | 0.2 M | 0 |
| 8 | NaN(SiMe$_3$)$_2$ (3.0) | Toluene | 0.2 M | 84 |
| 9 | LiO$^t$Bu (3.0) | Toluene | 0.2 M | 0 |
| 10 | NaO$^t$Bu (3.0) | Toluene | 0.2 M | 0 |
| 11 | KO$^t$Bu (3.0) | Toluene | 0.2 M | 0 |
| 12 | LiN(SiMe$_3$)$_2$ (3.0) | Toluene | 0.2 M | 23 |
| 13 | KN(SiMe$_3$)$_2$ (3.0) | Toluene | 0.2 M | 8 |
| 14 | NaN(SiMe$_3$)$_2$ (2.0) | Toluene | 0.2 M | 74 |
| 15 | NaN(SiMe$_3$)$_2$ (4.0) | Toluene | 0.2 M | 89 (86)[c] |
| 16[d] | NaN(SiMe$_3$)$_2$ (4.0) | Toluene | 0.2 M | 73 |
| 17[e] | NaN(SiMe$_3$)$_2$ (4.0) | Toluene | 0.2 M | 70 |
| 18 | NaN(SiMe$_3$)$_2$ (4.0) | Toluene | 0.1 M | 63 |

[a]Reaction conditions: **1a** (0.2 mmol, 2.0 equiv.), **2a** (0.1 mmol, 1.0 equiv.), room temperature, 12 h.
[b]Assay yields determined by $^1$H NMR spectroscopy of the crude reaction mixtures using CH$_2$Br$_2$ as an internal standard.
[c]Isolated yield.
[d]**1a** (1.5 equiv.).
[e]6 h.
*MTBE* methyl *tert*-butyl ether, *DME* dimethoxyethane, *CPME* cyclopentyl methyl ether, *THF* tetrahydrofuran, *DMSO* dimethyl sulfoxide, *DMF* N,N-dimethylformamide.

## Results

**Reaction optimization**. We initiated our reaction optimization using *N*-benzyl benzophenone imine **1a** and commercial allyl phenyl ether **2a** as coupling partners with 3.0 equiv. NaN(SiMe$_3$)$_2$ in MTBE (methyl *tert*-butyl ether) at room temperature for 12 h. To our delight, the allylation product **3aa** was generated in 64% assay yield (AY, as determined by $^1$H NMR integration against an internal standard Table 1, entry 1). We previously discovered that solvent can play an important role in modulating reactivity of 2-azaallyl anions by coordination to the main group cation of the base[69]. Therefore, a variety of solvents, including DME (1,2-dimethoxyethane), CPME (cyclopentyl methyl ether), THF (tetrahydrofuran), 1,4-dioxane, DMSO, DMF, and toluene, were examined (entries 2–8). CPME and toluene provided the target product **3aa** in 74% and 84% AY, respectively, while other solvents either gave reduced yields or led to no reaction. Using toluene, we next screened bases (LiO$^t$Bu, NaO$^t$Bu, KO$^t$Bu, LiN(SiMe$_3$)$_2$ and KN(SiMe$_3$)$_2$, entries 9–13). Of these, only LiN(SiMe$_3$)$_2$ and KN(SiMe$_3$)$_2$ afforded product **3aa** in 23% and 8% AY. Other bases did not result in the desired product. When 2.0 equiv. of NaN(SiMe$_3$)$_2$ (entry 14) was employed, the yield dropped to 74%. However, the yield increased to 89% (with 86% isolated yield) when 4.0 equiv. of NaN(SiMe$_3$)$_2$ (entry 15) was used. Further decreasing the amount of **1a** (from 2.0 to 1.5 equiv.), reaction time (from 12 h to 6 h), and concentration (from 0.2 M to 0.1 M) led to a decrease of yields to 63–73% (entries 16–18). Based on this optimization, the standard conditions for the allylic alkylation are those in entry 15 of Table 1.

Following the reaction optimization with allyl phenyl ether, we surveyed other allylic electrophiles. Under the optimized conditions, allyl substrates possessing leaving groups, such as allyl methyl ether

(18% AY), allyl benzyl ether (14% AY), allyl benzoate (0% AY), allyl acetate (19% AY), allyl bromide (66% AY), allyl *tert*-butyl silyl ether (16% AY), and allyl *tert*-butyl diphenylsilyl (14% AY) provided the desired product in lower AY than allyl phenyl ether (89% AY). Thus, we selected allyl phenyl ether **2a** as the allylating agent, which was easily synthesized from phenol.

**Reaction scope of imines**. With the optimized conditions in hand (Table 1, entry 15), we initiated investigation of the scope of *N*-benzyl ketimines **1** (Fig. 3a). In general, we found that a wide variety of ketimines with neutral, electron-rich, and electron-deficient Ar groups provided good to excellent yields. Electron-donating substituents 4-Me (**1b**) and 4-$^t$Bu (**1c**) generated allylic products **3ba** and **3ca** in 83% and 78% yields, respectively. 4-Methoxy and 3,4-methylenedioxy groups (**1d** and **1e**) delivered products **3da** and **3ea** in 63% and 70% yields, respectively. *N*-benzyl ketimines bearing electronegative and electron-withdrawing groups, such as 4-F, 4-Cl, 4-Br, and 3-CF$_3$, were also suitable coupling partners, providing the products **3fa**, **3ga**, **3ha**, and **3ia** in 73%, 45%, 35%, and 54% yields, respectively. Coupling with a ketimine possessing a biphenyl group (**1j**) produced the product **3ja** in 59% yield. The influence of more sterically hindered *N*-benzyl ketimines was explored. Interestingly, 1-naphthyl (**1k**) and 2-Tol (**1l**) ketimines reacted with the allyl ether in good yields (92% and 73%, respectively), despite the increased steric hindrance. Finally, the heterocyclic ketimine bearing a 3-pyridyl (**1m**) and 2-thiophenyl (**2n**) were also competent coupling partners, furnishing product **3ma** at 60 °C in 64% yield and **3na** in 32% yield. With imines bearing 4-C$_6$H$_4$-CN, 4-C$_6$H$_4$-COOMe, 2-pyridyl, 4-pyridyl, or 2-thiazolyl were employed, no reaction occurred and the allyl ethers were recovered in >90% yields. In addition to the allylation products in Fig. 3a, minor regioisomeric allylation products

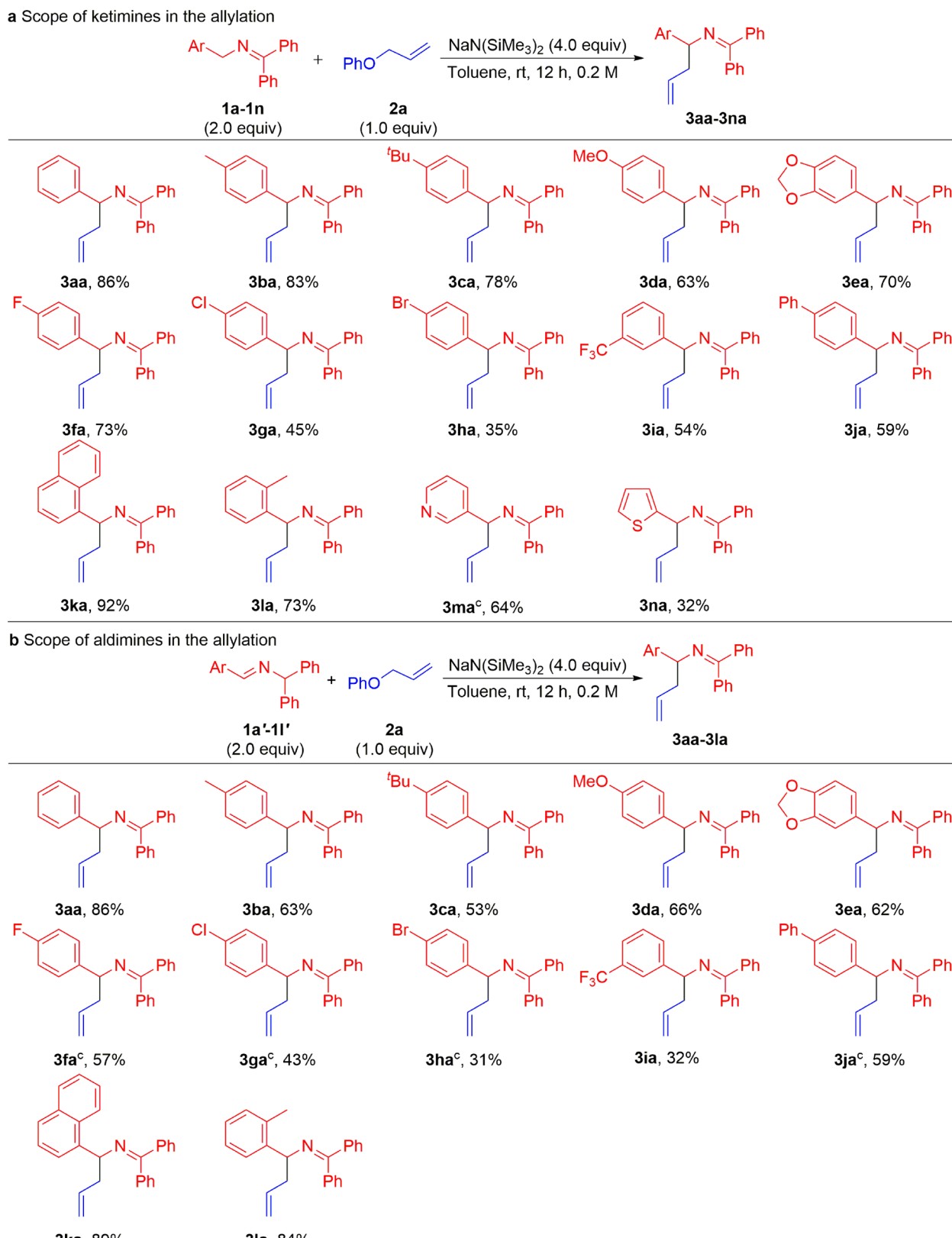

**Fig. 3 Substrate scope of imines[a,b]. a** Scope of ketimines in the allylation. **b** Scope of aldimines in the allylation. [a]Reactions were conducted on a 0.6 mmol scale using 2.0 equiv. ketimine, 1.0 equiv. **2a**, and 4.0 equiv. NaN(SiMe$_3$)$_2$ at 0.2 M. [b]Yield of isolated product after chromatographic purification. [c]At 60 °C.

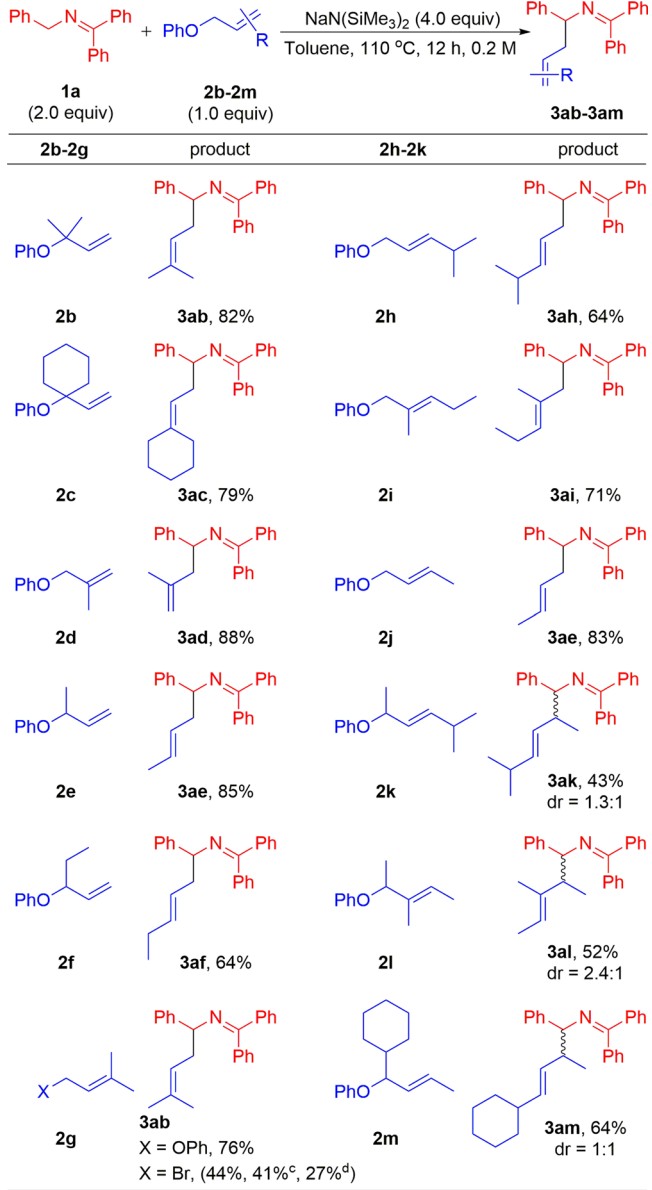

**Fig. 4 Substrate scope of allyl phenyl ethers[a,b].** [a]Reactions were conducted on a 0.6 mmol scale using 2.0 equiv. ketimine **1a**, 1.0 equiv. allyl ether, and 4.0 equiv. NaN(SiMe₃)₂ at 0.2 M and 110 °C. [b]Yield of isolated product after chromatographic purification. [c]3.0 equiv. NaN(SiMe₃)₂. [d]1.0 equiv. ketimine **1a**. dr, diastereomeric ratio.

**3aa′–3ma′** were detected in these reactions with yields ranging from 7% to 18% (see Supplementary information for details). This phenomenon is similar to the transition-metal-free arylation of 2-azaallyls reported previously by our group[63].

In an effort to fully explore the scope of this transformation, we next investigated aldimine substrates. In our past work, the aldimines have generally proven to be inferior starting materials to their ketimine isomers, despite generating the identical 2-azaallyl anions. This is attributed to the challenging deprotonation of the more hindered diphenylmethyl C–H bond. The advantage of aldimines, however, is that there are many commercially available benzaldehyde precursors[67]. As shown in Fig. 3b, the optimized conditions for ketimines accommodated aldimines bearing various substituted aryl groups in yields slightly below those reported in Fig. 3a. The parent aldimine (**1a′**) and those with Ar groups supporting alkyl substituents, such as 4-Me

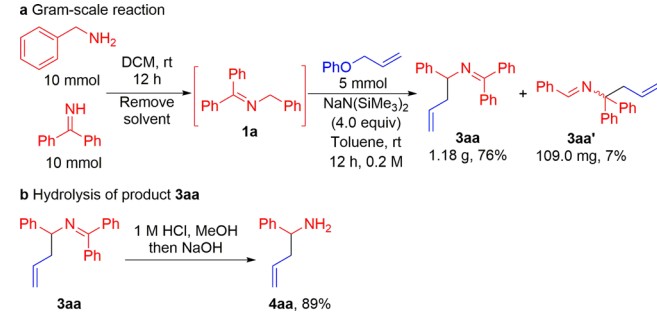

**Fig. 5 Synthetic applications. a** Gram-scale sequential one-pot imine generation/allylation process. **b** Allylated product hydrolysis to the homoallylic amine. DCM, dichloromethane.

(**1b′**) and 4-ᵗBu (**1c′**), furnished the desired products **3aa**, **3ba**, and **3ca** in 86%, 63%, and 53% yields, respectively. Aldimine substrates bearing electron-donating (4-OMe and dioxol), electronegative (4-F, 4-Cl, and 4-Br) and electron-withdrawing (3-CF₃) groups led to products **3da**, **3ea**, **3fa**, **3ga**, **3ha**, and **3ia** in 31–66% yields. Coupling with biphenyl substrate (**1j′**) proceeded in 59% yield. The sterically hindered 1-naphthyl and 2-Me derivatives reacted with allyl phenyl ether to form the desired products **3ka** and **3la** in 89% and 84% yields. The higher yield of the aldimine **1l′** over its ketimine counterpart **1l** may be due to the increased steric hindrance about the benzylic C–H's of the ketimine **1l**, causing a lower conversion to the 2-azaallyl anion and subsequent yield of the products. Similarly, a few regioisomeric allylation products with yields ranging from 6% to 23% were obtained when aldimines were used as 2-azaallyl anion sources (see Supplementary information for details).

Encouraged by the results with ketimine and aldimine substrates, we turned our attention to surveying the scope in the allyl phenyl ether-coupling partner. Although the substrates examined were more sterically hindered and required higher temperature (110 °C), the scope was found to be broad (Fig. 4). Mono- or 1,1-disubstituted allyl ethers, such as 2-methylbut-3-en-2-yl (**2b**), 1-vinylcyclohexyl (**2c**), 2-methylallyl (**2d**), but-3-en-2-yl (**2e**), and pent-1-en-3-yl (**2f**) groups, were coupled with N-benzyl ketimine **1a** at the least hindered position of the allyl ether to generate coupling products **3ab**, **3ac**, **3ad**, **3ae**, and **3af** in 82%, 79%, 88%, 85%, and 64% yields, respectively. 1,2-Disubstituted allyl ethers, such as 3-methylbut-2-en-1-yl (**2g**), E-4-methylpent-2-en-1-yl (**2h**), E-2-methylpent-2-en-1-yl (**2i**), and E-but-2-en-1-yl (**2j**) groups, furnished linear coupling products **3ab**, **3ah**, **3ai**, and **3ae** in 76%, 64%, 71%, and 83% yields, respectively. Interestingly, the 1,2-disubstituted allyl bromide 3-methylbut-2-en-1-yl (**2g′**) provided product **3ab** in a lower yield (44%) compared to the phenolic electrophile (76% yield). Allyl ethers with substituents both on the terminal and allylic positions, such as (E)-5-methylhex-3-en-2-yl (**2k**), (E)-3-methylpent-3-en-2-yl (**2l**), and (E)-1-cyclohexylbut-2-en-1-yl (**2m**) provided the less sterically hindered coupling products **3ak**, **3al**, and **3am** in 43%, 52%, and 64% yields, respectively. The moderate yields may be due to increased steric hindrance in the C–C bond-forming step. In addition, regioisomeric allylation products **3ab′** and **3ah′** were obtained with yields in 7% and 23%, respectively (see Supplementary Information for details). Notably, deprotonative isomerization or cyclization of allylated products was not detected for any of the coupling reactions in Figs. 3 and 4.

It is vital for a synthetic approach to be straightforward and scalable. Hence, we explored the scalability of this coupling reaction by a telescoped imine preparation/allylation process on gram scale (Fig. 5a). Treatment of the benzyl amine with

benzophenone imine in DCM at room temperature for 12 h was followed by removal of the solvent under reduced pressure to form *N*-benzyl ketimine **1a**. Next, the unpurified **1a** was coupled with allyl phenyl ether **2a** following the standard procedure. After 12 h at room temperature, workup, and purification 1.18 g of **3aa** (76% over 2 steps) and **3aa′** (109.0 mg, 7%) were generated. Hydrolysis of the allylated product **3aa** was performed to deliver the homoallylic amine **4aa** in 89% yield (Fig. 5b).

**Mechanistic studies**. To obtain insight into the allylation reaction pathway, we carried out preliminary mechanistic investigations. In order to isolate the leaving group, we switched to an allyl aryl ether that would generate a less volatile phenolic product. When an allyl ether bearing a 2-naphthyl group **5a** was employed for coupling with *N*-benzyl ketimine **1a**, naphthalen-2-ol **6a** was isolated in 90% yield together with the allylated products **3aa** and **3aa′** in 82% and 7% yields, respectively. This result indicates that 2-napthoxide was generated under the reaction conditions, followed by protonation upon workup (Fig. 6a).

To further probe the mechanism of the allylation reaction, a Hammett study was performed using intermolecular competition experiments (Fig. 6b). At the outset of these experiments, we were aware that the correlation of the relative rates might be impacted by concurrent reaction mechanisms ($S_N2$, $S_N2'$, and radical). In the event, the Hammett plots show a loose correlation with typical polar substituent constant parameters ($R^2 = 0.55$ for $\sigma$, $R^2 = 0.53$ for $\sigma^-$). The fit was improved by employing the $\sigma^\bullet$ parameter ($R^2 = 0.62$) (see Supplementary information for details). The experimental data could be better fitted to a two-parameter Hammett relationship[104–109]. For example, plotting $\log(k_{rel})$ versus a combination of $\sigma$ (33%) and $\sigma^\bullet$ (67%) provided a better fit ($R^2 = 0.70$) (Fig. 6c), which reflects the character of the selectivity-determining and rate-limiting step that might be expected from a combination of radical character with polar influences. The $\rho$ value determined ($\rho = +1.4$), with this combination of $\sigma$ scales, is smaller than would be expected for a SET mechanism, though still consistent with the buildup of negative charge[105].

To probe the presence of radical intermediates in the allylation reaction, two radical-clock-containing cyclopropanes were prepared (see Supplementary Information, Synthesis of radical clock **7a** and **11a** for details). In the case of allylic ether radical clock **7a**,

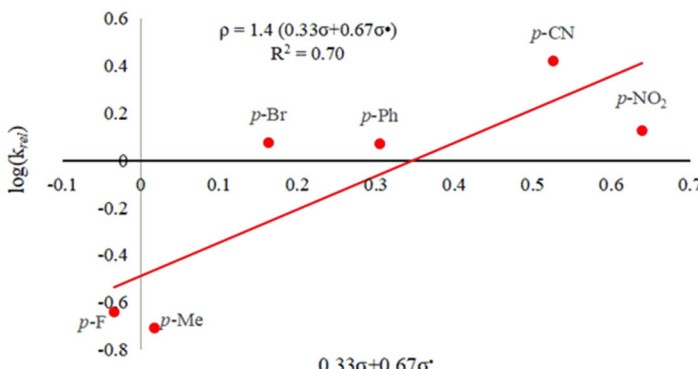

**a** Isolation of the leaving group

**b** Competition experiments

**c** Hammett plot of experimental log($k_{rel}$) *vs.* calculated 0.33σ+0.67σ•

**Fig. 6 Mechanistic probes. a** Isolation of the leaving group. **b** Competition experiments. **c** Plot of log($k_{rel}$) versus a combination of σ (33%) and σ• (67%). Scatter in the plot is likely due to mixed mechanisms.

the $S_N2$ and $S_N2'$ reaction pathways are hindered by the bulky substituents. The reaction of the cyclopropane radical clock **7a** (2.0 mmol) with ketimine **1a** in the presence of NaN(SiMe₃)₂ provided the allylated product **8aa** in 32% and the cyclopropane ring-opened product **9aa** in 15% yield (Fig. 7a). It is noteworthy that radical clock **7a** is expected to favor ring-closed products, because ring opening produces a high-energy primary radical (Fig. 7a). Nonetheless, these results suggest that the coupling of hindered allyl phenyl ethers proceed, at least in part, through radical intermediates. A control experiment was carried out with **7a** at 110 °C and NaN(SiMe₃)₂, but in the absence of ketimine **1a**. Only the Claisen rearrangement product **10a** was obtained (96% yield, Fig. 7b). No ring-opened product was observed. This result suggests that NaN(SiMe₃)₂ alone is not reacting as a reducing agent and the ketimine is necessary to generate radical intermediates.

The radical clock **11a** was designed with a terminal double bond to facilitate the $S_N2'$ reaction and with a phenyl cyclopropyl moiety that would give a benzylic radical if this substrate proceeded through radical ring-opened intermediates. When **11a** (2.0 mmol) was subjected to ketimine **1a** and NaN(SiMe₃)₂ at 110 °C, the allylated product **12aa** was afforded in 83% yield with the cyclopropane intact. This observation suggests that, in the case of unhindered pathways for $S_N2'$, the two-electron process prevails. The cyclopropane ring-opened product **13aa** was isolated in 15% yield (Fig. 8a). Crystals were obtained of the minor product **13aa** and the structure confirmed by X-ray crystallography (CCDC 2039076).

A proposed mechanism for the formation of **13aa** is provided in Fig. 8b. Based on DFT calculations, we previously proposed that the 2-azaallyl anion (**S1**) could undergo SET with ketimine **1a** to generate the 2-azaallyl radical (**S3**) and the ketiminyl radical anion (**S2**). Here, the resulting 2-azaallyl radical (**S3**) undergoes addition to the double bond to generate a C–C bond and a new radical (**S4**). This radical can eliminate the phenoxy radical, which can abstract H• from the benzylic hydrogen in **S4**. Formation of phenol is accompanied by generation of radical **S5** and its resonance form **S6**. The radical character in **S6** can add to the π-system of the newly formed double bond to generate **S7**. This addition places the radical alpha to the cyclopropyl group, which opens to give the stabilized benzylic radical **S8**. Intermediate **S8** then gains a hydrogen and an electron, possibly through reduction of the benzylic radical by the ketiminyl anion (**S2**) and proton transfer from HN(SiMe₃)₂. This mechanism is reminiscent of our previous work[85], in which vinyl bromides reacted with 2-azaallyl species via either an anionic substitution pathway with the 2-azaallyl anion or a radical pathway with the 2-azaallyl radical.

In order to probe this system for radical behavior, an additional set of experiments were performed (Fig. 9a). First, the allylation product **3aa** (0.4 mmol) was reacted in the presence of NaN(SiMe₃)₂ (4.0 equiv.) for 0.5 h at room temperature (Fig. 9a). This resulted in the formation of a dark purple solution, consistent with deprotonation of the ketimine to generate the 2-azaallyl anion. Next, ketimine **1a** was added at 110 °C. After heating at this temperature for 12 h, the reaction was worked up following the standard procedure. Dihydropyrrole product **14aa** was isolated in 38% yield. A proposed mechanism for the formation of **14aa** is provided that is based on the notion that the 2-azaallyl anion can undergo SET with ketimine **1a** to generate the ketiminyl anion and 2-azaallyl radical, as we previously reported[85]. One could imagine that the ketimines and aldimines are in equilibrium with the 2-azaallyl anions. Under such circumstances, SET from 2-azaallyl anion **S9** to ketimine **1a** would generate ketiminyl anion **S2** and the 2-azaallyl radical (**S10**). Resonance form **S11** could then undergo addition to the

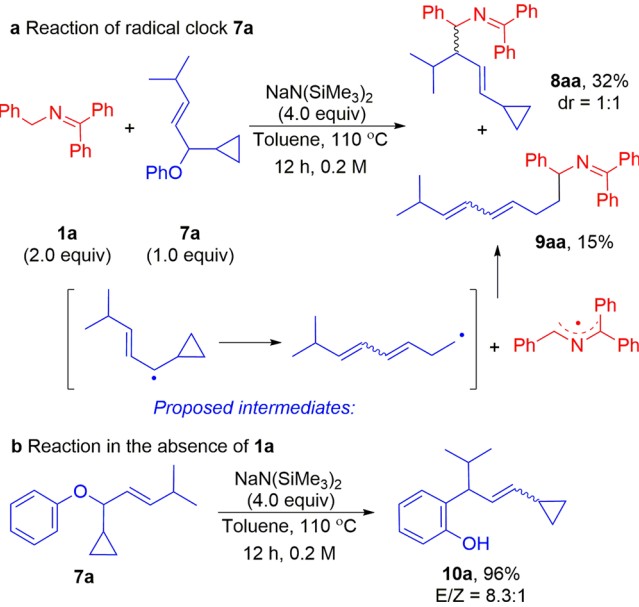

**Fig. 7 Mechanistic experiments. a** Radical clock study with **7a**. **b** Control experiment in the absence of ketimine **1a**.

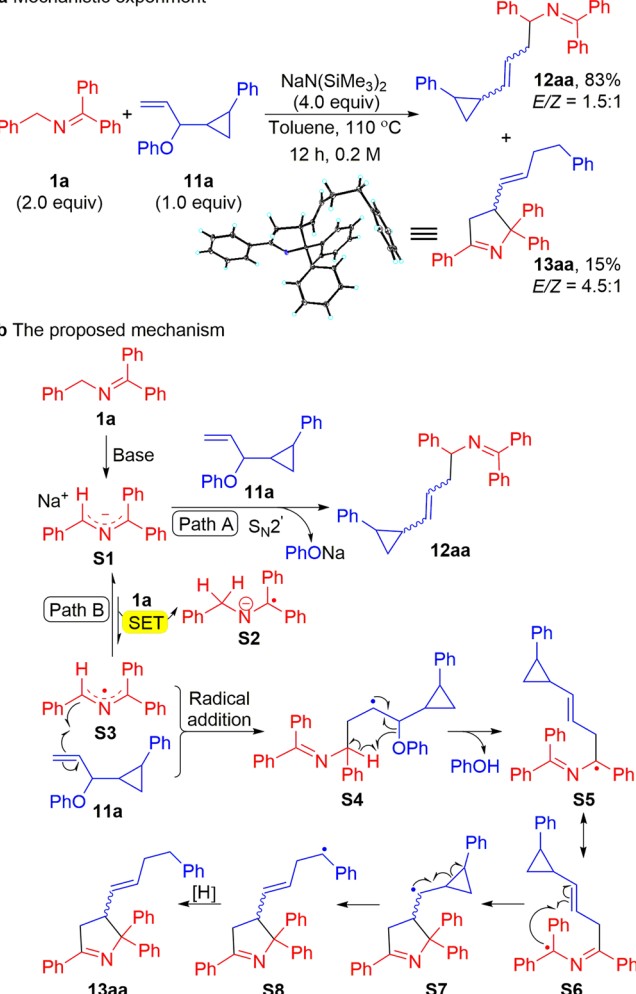

**Fig. 8 Mechanistic studies. a** Reaction of radical clock **11a** with in situ generated 2-azaallyl anion. **b** The proposed mechanism for the formation of **13aa**. SET, single electron transfer.

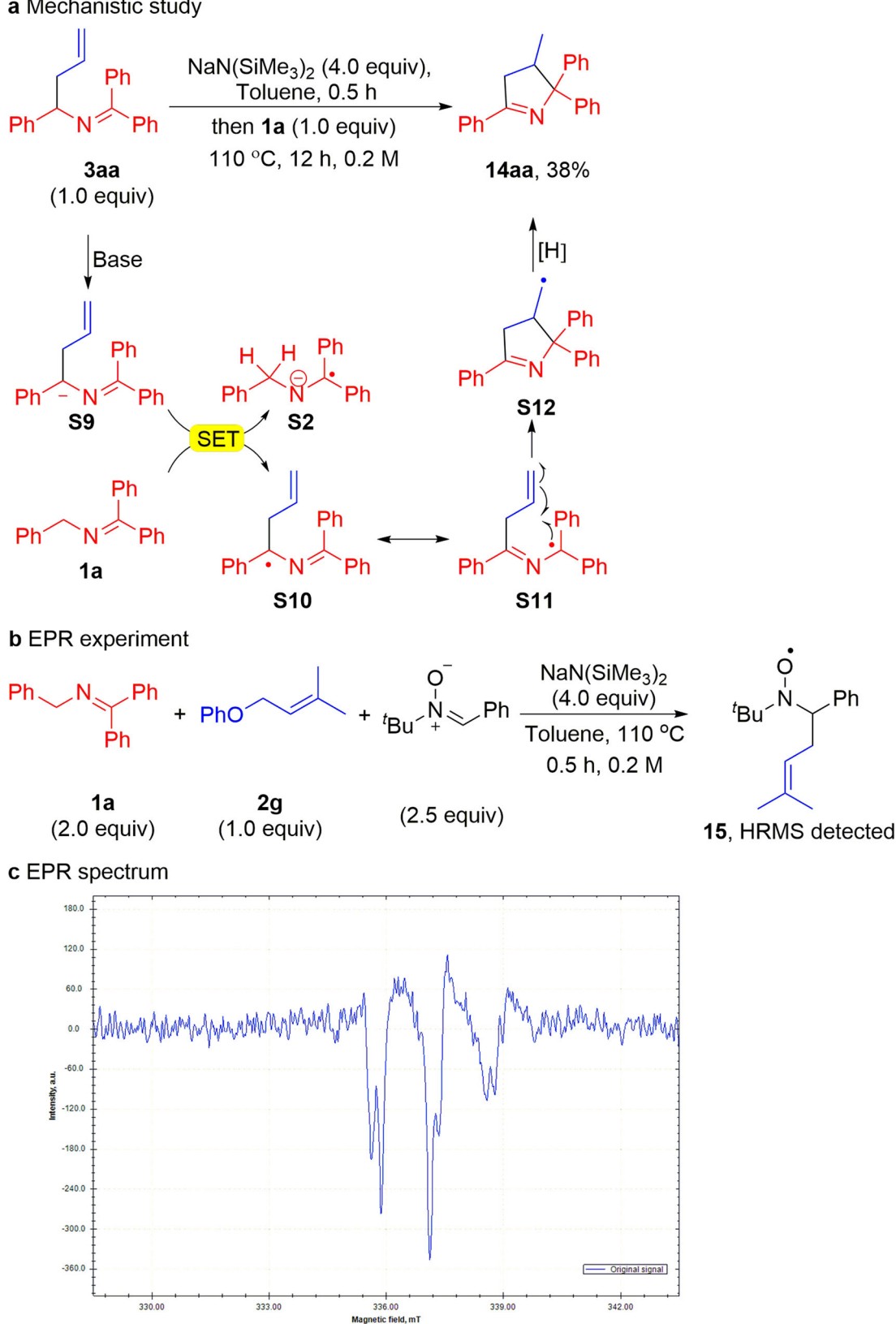

**Fig. 9 Mechanistic and EPR studies. a** Mechanistic experiment with allylation product **3aa**. **b** EPR experimental conditions. **c** X-band EPR spectrum of the PBN-trapped carbon-centered radical ($T = 298$ K; microwave frequency: 9.462390 GHz; power: 0.2 mW; center field: 336.00 mT; sweep width: 15.0 mT; modulation frequency, 100 kHz; modulation amplitude, 100 μT). SET, single electron transfer. EPR, electron paramagnetic resonance.

double bond to generate a C–C bond and a new radical (**S12**). Intermediate **S12** gains a hydrogen and an electron, possibly via HAT from **S2** to form the dihydropyrrole **14aa**. While the exact mechanism of this transformation is not clear, it does have the hallmarks of a radical process rather than a two-electron addition of the 2-azaallyl anion to the double bond, which would give rise to a primary carbanion. When the reaction is carried out by combining both **3aa** and **1a** at the same time, the yield of the dihydropyrrole increased to 47%. We note that when the reaction is conducted in the absence of **1a**, no major product is observed and less than 5% **14aa** is detected by NMR.

Spin trapping experiments using phenyl *N-tert*-butylnitrone (PBN) as the spin trap support the proposed radical-type mechanism. Heating a mixture of **1a**, **2g**, NaN(SiMe$_3$)$_2$ in the presence of PBN led to the formation of a PBN-trapped carbon-centered radical, as detected by EPR spectroscopy (Fig. 9b, c). The resulting EPR signal ($g = 2.0040$, $A_N = 14.9$ G, $A_H = 2.4$ G) is strong and similar to other reported PBN-trapped carbon-centered radicals[110,111]. The cationic signal of radical **15** can be detected in the reaction mixtures by high-resolution mass spectroscopy (HRMS calculated for C$_{16}$H$_{24}$NO$^{\bullet+}$ 246.1852, found 246.1851 [M]$^{\bullet+}$).

## Discussion

We have outlined reactivity of 2-azaallyl anions that is founded in their ability to behave as super electron donors. This chemistry represents a unique transition-metal-free allylation of 2-azaallyls with allyl ethers to prepare homoallylic amine derivatives, which are of value in the pharmaceutical industry. In this reaction, simple, readily prepared allyl phenyl ethers coupled with azaallyl anions or azaallyl radicals to construct new C(sp$^3$)–C(sp$^3$) bonds in excellent yields. Notably, the simple combination of base and solvent enabled the metal-free allylation to proceed efficiently, in which no deprotonative isomerization or cyclization of products was detected. A gram-scale telescoped homoallylic amine preparation was carried out, demonstrating the potential synthetic utility of this chemistry. In addition, mechanistic studies provide insight into these C(sp$^3$)–C(sp$^3$) bond-forming reactions and support substrate-dependent radical and anionic pathways. Unlike past advances, this allylation approach enables the synthesis of a diverse array of homoallylic amines without the addition of transition-metal catalysts, photocatalysts, or organo-metallic reagents. These attributes increase the attractiveness of this method for applications in the pharmaceutical industry[112].

## Methods

**General procedure**. An oven-dried 8 mL reaction vial equipped with a stir bar was charged with ketimine **1** (1.2 mmol) or aldimine **1′** (1.2 mmol) and allyl phenyl ether **2** (0.6 mmol) under a nitrogen atmosphere in a glove box. A solution of NaN(SiMe$_3$)$_2$ (2.4 mmol) in 3 mL dry toluene was added to the reaction vial. The reaction mixture turned to a dark purple solution. Then the vial was sealed with a cap, removed from the glove box, and stirred for 12 h at room temperature (Fig. 3) or 110 °C (Fig. 4). The room temperature reaction mixture was opened to air, quenched with three drops of H$_2$O, diluted with 3 mL of ethyl acetate, and filtered over a 2 cm pad of MgSO$_4$ and deactivated silica. The pad was rinsed with ethyl acetate (3 × 2 mL), and the combined organic solutions were concentrated in vacuo. The crude material was purified on an Agilent HPLC 1260 system using acetonitrile:H$_2$O (75:25 vol./vol.) as the mobile phase and flow rate of 3.5 mL/min with monitoring at 254 nm to give product **3**.

## Data availability

The authors declare that the data supporting the findings of this study are available within the article and its Supplementary information files. For the experimental procedures and spectroscopic and physical data of compounds, see Supplementary Methods. For $^1$H and $^{13}$C{$^1$H} NMR spectra of compounds, see Supplementary Figs. 1–127. The X-ray crystallographic coordinates for structures reported in this study have been deposited at the Cambridge Crystallographic Data Centre (CCDC), under deposition numbers of CCDC 2039076. These data can be obtained free of charge from The Cambridge Crystallographic Data Centre via www.ccdc.cam.ac.uk/data_request/cif.

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

## Acknowledgements

This work was supported by grants from the National Key Research and Development Program of China (2019YFE0109200), NSFC (U1702286), Program for Changjiang Scholars and Innovative Research Teams in Universities (IRT17R94), the China Post-doctoral Science Foundation (2019M663581), the NSF of Yunnan Province (2019FY003010 and 202005AB160003), the YunLing Scholar Programs and IRTSTYN. P.J.W. thanks the US National Science Foundation (CHE-1902509) for financial support. We thank Prof. Chengfeng Xia for the help with EPR equipments.

## Author contributions

G.D. and S.D. contributed equally to this work. X.Y. conceived of the project. H.Z. and P.J.W. designed the experiments. G.D., S.D., J.W., Z.C., T.L., and W.C. performed the research. X.Y. and P.J.W. wrote the manuscript.

## Competing interests

The authors declare no competing interests.
