## [Peer Review File · Nature Communications]

Reviewers' Comments:

Reviewer #1:

Remarks to the Author:

The submitted article entitled "Transition-metal-free allylation of 2-azaallyls with allyl ethers through polar and radical mechanisms" represents a mechanistically interesting advance from the collaborative efforts of the Zhang, Yang, and Walsh groups that builds off of previous discoveries from their team and others. From a procedural standpoint, the authors determined that homoallylic amines, or more specifically the benzophenone imines of homoallylic amines, can be readily constructed from the condensation of base-generated 2-azaallyl anions with phenol allyl ethers. Preliminary mechanistic studies suggest that both polar (2-electron) and radical mechanisms successfully compete with each other to produce the observed products.

It has been known since Kauffmann's studies in 1977 (T. Kauffmann et al. Chem. Ber. 1977, 110, 2659-2664) that allyl bromide is a sufficient electrophile for the allylation of 2-azaallyl anions (surprisingly, Kauffmann's seminal contributions are not cited in this submission). Additionally, as the authors rightly note, there are myriad other strategies for the construction of homoallylic amines, several of which are far superior to the procedure in question in regard to yield, substrate scope, and stereoselectivity. While the described synthetic procedures are not likely to supplant current art, they could offer significant fundamental mechanistic insights likely to influence thinking in the field. Unfortunately, the mechanistic studies and corresponding reaction scope(s) presented in the submitted article are too underpowered to merit publication as is in Nature Communications. If the authors address the concerns listed below, however, it is very conceivable that the resulting manuscript would rise up to the acceptable standard.

MAJOR ISSUES TO ADDRESS:

1. Overall, the analysis of the reaction products is insufficient:

1A. The authors note on several occasions that "deprotonative isomerization or cyclization of allylated products" were not detected, but what about the other regioisomer in which the more substituted diphenylmethine carbon was allylated? Several studies on the allylation and arylation of 2-azaallyl anions and radicals (including those from the authors of this submitted manuscript) note that a mixture of regioisomers, always favoring the major products noted in this paper, are produced. Given the authors' familiarity with this phenomenon, it is surprising that it is not mentioned or explored within the manuscript. Indeed, if the minor regioisomeric allylation product was never detected, that would represent a major breakthrough in the field requiring significantly more mechanistic explanation and fanfare.

1B. Does any of the amide (H₂N⁻) base get allylated under the reaction conditions to form the corresponding allyl amine? Such a process could explain why allyl bromide (a stronger electrophile) and allyl benzoate (with an electrophilic ester moiety) were inferior to phenyl allyl ether under reaction conditions in which four equivalents (!) of the nucleophilic base are used. Similarly, this (competitive prenylation of NH₂) could explain why 3ab is formed in only 44% with prenyl bromide under the reaction conditions. If the amount of base and 2-azaallyl anion were decreased, does the yield for prenylation with prenyl bromide increase?

1C. Several of these questions could be answered by the reviewer if the authors had included at least one example of the ¹H NMR spectrum of the crude reaction mixture(s) employed to determine the assay yields (with CH₂Br₂ as an internal standard). At the end of the paper the "authors declare that the data supporting the findings of this study are available within the article and its Supplementary Information files," but this is not an accurate statement given that the crude NMR used to determine assay yields are not available. The authors also did not report the specific procedures for how they determined the assay yields. Did they run the ¹H NMR analysis with the CH₂Br₂ standard before or after HPLC purification? If the latter, then this is insufficient as HPLC purification will remove any evidence of other products in the reaction. Analysis of the crude reaction mixture is absolutely required to provide a clearer picture of the reaction.

2. The substrate scopes presented in Tables 2 and 3 are incomplete. There is only one example of an "electron-withdrawing" substituent on the 2-azaallyl framework in either table and that is the meta-CF₃ group. The para-fluoride is not truly electron-withdrawing (the authors deftly describe it as electronegative), because the Hammett resonance coefficient for a para-F is slightly negative. The meta-CF₃ group, while definitely electron-withdrawing, is not in resonance with the

delocalized 2-azaallyl anion, so its impact is expected to be negligible. The authors must include at least one example of a "true" electron-withdrawing group on the para position. Additionally, it would be helpful if that group was also a competitive electrophile, such as a nitrile or ester, to explore the full substrate scope. If those substrates don't work, this needs to be reported. Another substrate that should be included in Tables 2 and 3 is a heteroaromatic ring in which the anion could delocalize onto the heteroatom (e.g. 2-pyridyl or 2-thiazolyl). The authors do include a 3-pyridyl, which is good, but this does not really challenge the system.

3. The solvent screening studies are also incomplete. Several studies, including those involving 2-azaallyl anions/radicals, indicate that DMSO and DMF are superior (albeit not innocent) solvents for electron-transfer processes. Results from these two solvents should be included in Table 1.

4. The "how" of the reaction mechanism is underexplored, particularly the role of the phenol/phenoxide leaving group. In Figure 5 and the corresponding text, the authors note that 2-naphthol is isolated in 90% yield when allyl 2-naphthyl ether (5a) is used in the reaction. This is an encouraging result that should be explored further to gain a better mechanistic insight. For example, a small collection of para-substituted phenols should be employed as substrates for the reaction and the corresponding reaction rates monitored to construct a Hammett plot analysis. The Hammett plots should follow different constants (anion resonance constant or radical constant) based on the substituents whether the phenolic group leaves as an anion (phenoxide) or radical.

5. The results with allyl ethers 2e and 2j are interesting but underexplored. In both cases, alkylation occurs at the less substituted carbon of the allylic framework to provide homoallylic imine 3ae. But it is unclear if 3ae is formed from 2e by an SN2' reaction mechanism or if something else is at play. Judicious use of a ¹³C- or ²H-labelled 2e and 2j would provide useful insight on the likelihood/predominance of SN2' and SN2 reaction mechanisms for the respective allylations to form the same product.

OTHER ISSUES TO ADDRESS:

6. In the opening sentence, the authors quote the "escaping from the flatlands" concept in medicinal chemistry, but forget to cite the two papers from Lovering that codified that concept. These papers must be cited: F. Lovering, J. Bikker, and C. Humblet, *J. Med. Chem.* 2009, 52(21), 6752-6756 & F. Lovering, *MedChemComm*, 2013, 4, 515-519.

7. At the beginning of the second paragraph, the authors state that "homoallylic amines are found in a vast number of biologically active molecules," but it is not clear how accurate this statement is. Indeed, homoallylic amines are incredibly useful precursors for the synthesis of a vast number of biologically active molecules. But, to the best of my knowledge, there are very few examples (if any) of biologically active molecules that are just homoallylic amines similar to the one synthesized at gram-scale in this article (4aa).

8. At the end of the second paragraph, the authors forget to mention that Ni catalysis has also been used to synthesize homoallylic amines (C. Liu et al. *J. Org. Chem.* 2019, 84, 10102-10110). This paper was cited in the Supporting Information, but it is appropriate to mention it here, as well.

9. In the introduction, there are several examples of verb tenses not agreeing, which leads to some confusion. For example on page 3: "...we *demonstrated* that 2-azaallyl anions *serve* as SEDs and *enabled* transition metal-free..."

10. At the end of the paragraph before Table 3, the authors write "the higher yield of the aldimine 1j' over its ketimine counterpart 1j maybe due to the increased steric hindrance about the benzylic C-H's of the ketimine." This explanation is not very convincing. Why should it be so much different for 1j' versus 1j if the reactive nucleophile/radical in the reaction is the exact same delocalized species? It is possible that aldimine 1j' is less susceptible to hydrolysis than benzophenone imine 1j, so the reagent survives the presence of adventitious water better. Is this what they mean?

Reviewer #2:

Remarks to the Author:

The entitled manuscript "Transition-metal-free allylation of 2-azaallyls with allyl ethers through polar and radical mechanisms" disclosed the novel reactivity of 2-azaallyl anions with allyl ethers under transition-metal-free conditions. The present transformation exhibits excellent substrate generality with respect to both the coupling partners to prepare homoallylic amine derivatives in good to excellent yields. Of note, straightforward gram scale preparation and simple combination of base and solvent demonstrated potential synthetic utility and ease of operation of this method. Reasonable design of mechanistic studies provided the polar and radical mechanisms process. Overall the present method is a notable addition to the chemistry of allylic alkylations and certainly satisfies both novelty and quality criteria of Nat. Commun.. The supporting information contains all the necessary experimental procedures and compounds characterization.

Some suggestions and questions for improvement of the manuscript before publication are given here below:

- 1) In the maintext, "see Supporting Information, 'Synthesis of radical clock 7a and 11a' for details", "11a" should be "11a"; "using acetonitrile:H₂O (75:25 vol./vol.) the as mobile phase", "as the mobile phase"?
- 2) Although the reaction of 1a with 2a and 5a showed no obvious differences, aromatic rings with electronic abundant or deficient substituents should be discussed.
- 3) How to determine the double bond geometric configurations of 3ae, 3af and other similar compounds?
- 4) Light free control experiment may be needed to exclude the influence of ambient light source, such CFL promoted SET process via electron donor acceptor complexes between the coupling partners.

Reviewer #3:

Remarks to the Author:

In this manuscript, Zhang and co-workers described a transition-metal-free allylation of 2-azaallyls with allyl ethers electrophiles. Although this reaction gave homoallylic amine derivatives with generally good yields, there are still some features that slightly reduce the importance of this contribution. First of all, its novelty is not high enough. The C-H bond adjacent to the imine nitrogen can undergo deprotonation to form 2-azaallyl anions with strong bases, and subsequent allylation with electrophiles is a common work, similar approach has already been reported for decades. The unique radical mechanism suggested in this manuscript only applies to some specific substrates. In fact, since the selectivity was quite good, no double bond migration isomer products were obtained (especially for 2g-2j), two electron S_N2 process is more likely. Moreover, The manuscript was not well organized. It will be better if more discussion were added, but not simply replicate the content of tables, especially in reaction optimization and substrate scope investigation section. In addition, the functional group tolerance was limited with respect to 2-azaallyls and high temperature was required for expansion of allyl phenyl ethers. Overall, the referee believes the current manuscript is not suitable for Nature Communication. More comments are listed below:

- (1) The authors should consider using radical scavengers such as TEMPO to probe the radical mechanism.
- (2) For the products containing F atom, the ¹⁹F NMR is normally required in the SI, the author should provide the data of ¹⁹F NMR for the compounds 3fa, 3ga.

Manuscript: "Transition-metal-free allylation of 2-azaallyls with allyl ethers through polar and radical mechanisms" (Manuscript ID: NCOMMS-20-45659) by Guogang Deng, Shengzu Duan, Jing Wang, Zhuo Chen, Tongqi Liu, Wen Chen, Hongbin Zhang, Xiaodong Yang and me.

The comments of the reviewers are valuable and very helpful for revising and improving our paper. We have studied comments carefully and performed many experiments and revisions. The full comments of the reviewers are pasted below and the corrections in the paper and the responds to the Reviewers' comments are as flowing:

Responds to the Reviewers' comments:

Referee: 1

Comments:

The submitted article entitled "Transition-metal-free allylation of 2-azaallyls with allyl ethers through polar and radical mechanisms" represents a mechanistically interesting advance from the collaborative efforts of the Zhang, Yang, and Walsh groups that builds off of previous discoveries from their team and others. From a procedural standpoint, the authors determined that homoallylic amines, or more specifically the benzophenone imines of homoallylic amines, can be readily constructed from the condensation of base-generated 2-azaallyl anions with phenol allyl ethers. Preliminary mechanistic studies suggest that both polar (2-electron) and radical mechanisms successfully compete with each other to produce the observed products.

It has been known since Kauffmann's studies in 1977 (T. Kauffmann et al. Chem. Ber. 1977, 110, 2659-2664) that allyl bromide is a sufficient electrophile for the allylation of 2-azaallyl anions (surprisingly, Kauffmann's seminal contributions are not cited in this submission).

Response:

We thank the reviewer for all their insightful comments. In the revised manuscript, we have cited the Kauffmann's seminal work (**Ref. 34:** T. Kauffmann et al. *Chem. Ber.* **110**, 2659–2664 (1977)). We have also highlighted Kauffmann's pioneering work of homoallylic amines synthesis as: "In particular Kauffmann and co-workers have prepared homoallylic amines by allylation of 2-azaallyl anions using allyl bromide as an electrophile." in the revised manuscript (Page 2).

Additionally, as the authors rightly note, there are myriad other strategies for the construction of homoallylic amines, several of which are far superior to the procedure in question in regard to yield, substrate scope, and stereoselectivity. While the described synthetic procedures are not likely to supplant current art, they could offer significant fundamental mechanistic insights likely to influence thinking in the field.

Response:

We agree with the reviewer on this point. Our work complements known methods. It is most likely to find applications when practitioners are interested in using electrophilic partners that are stable to most reaction conditions. We also agree that the conceptual novelty and fundamental mechanistic insights described in this study may have the most significant impact.

Unfortunately, the mechanistic studies and corresponding reaction scope(s) presented in the submitted article are too underpowered to merit publication as is in Nature Communications. If the authors address the concerns listed below, however, it is very conceivable that the resulting manuscript would rise up to the acceptable standard.

MAJOR ISSUES TO ADDRESS:

1. Overall, the analysis of the reaction products is insufficient:

1A. The authors note on several occasions that "deprotonative isomerization or cyclization of allylated products" were not detected, but what about the other regioisomer in which the more substituted diphenylmethine carbon was allylated? Several studies on the allylation and arylation of 2-azaallyl anions and radicals (including those from the authors of this submitted manuscript) note that a mixture of regioisomers, always favoring the major products noted in this paper, are produced. Given the authors' familiarity with this phenomenon, it is surprising that it is not mentioned or explored within the manuscript. Indeed, if the minor regioisomeric allylation product was never detected, that would represent a major breakthrough in the field requiring significantly more mechanistic explanation and fanfare.

Response:

Deprotonative isomerization or cyclization of allylated products was not detected for any of the coupling reactions in Tables 2–4. For the other regioisomer, in which the more substituted diphenylmethine carbon was allylated, among the 38 examples, the regioisomers of 17 examples were not detected. For the other 21 examples, we detected the minor regioisomeric allylation products. However, the yields were relatively low. We have added the results of minor regioisomer in the Supplementary Information.

According to reviewer's suggestion, we have added the results in the revised manuscript as: "In addition, several minor regioisomeric allylation products **3aa'**-**3ma'**, **3ab'** and **3ah'** were detected in these reactions with yields ranging from 6% to 23% (see Supplementary Information for details).

*1B. Does any of the amide (H₂N-) base get allylated under the reaction conditions to form the corresponding allyl amine? Such a process could explain why allyl bromide (a stronger electrophile) and allyl benzoate (with an electrophilic ester moiety) were inferior to phenyl allyl ether under reaction conditions in which four equivalents (!) of the nucleophilic base are used. Similarly, this (competitive prenylation of NH₂) could explain why **3ab** is formed in only 44% with prenyl bromide under the reaction conditions. If the amount of base and 2-azaallyl anion were decreased, does the yield for prenylation with prenyl bromide increase?*

Response:

The MN(SiMe₃)₂ did not get allylated under the reaction conditions to form the corresponding allyl amine. We have also tested the reaction of phenylmethanamine and phenyl allyl ether shown below. However, no reaction was observed under the reaction conditions.

As shown in the table below, we decreased the amount of base and 2-azaallyl anion for prenylation with prenyl bromide. Reducing the equivalent of base from 4 to 3, 2 or 1 led to a drop in the yields to 41, 38 or 29%, respectively (entries 1-3). Similarly, reducing the equivalent of ketimine **1a** from 2.0 to 1.0 led to a significant drop in yield to 27, 25, 22 or 8%, respectively (entries 4-7). We interpret the results as adequate concentration of azaallyl anion is needed for efficient addition of electrophiles. We have added the results in Table 4 (notes [c] and [d]) in the revised manuscript.

Entry	1a (equiv)	NaN(SiMe ₃) ₂ (equiv)	Assay yield of 3ab (%) ^a
1	2.0	3.0	41
2	2.0	2.0	38
3	2.0	1.0	29
4	1.0	4.0	27
5	1.0	3.0	25
6	1.0	2.0	22
7	1.0	1.0	8

^aAssay yields determined by ¹H NMR spectroscopy of the crude reaction mixtures using CH₂Br₂ as an internal standard.

1C. Several of these questions could be answered by the reviewer if the authors had included at least one example of the ¹H NMR spectrum of the crude reaction mixture(s) employed to determine the assay yields (with CH₂Br₂ as an internal standard). At the end of the paper the "authors declare that the data supporting the findings of this study are available within the article and its Supplementary Information files," but this is not an accurate statement given that the crude NMR used to determine assay yields are not available. The authors also did not report the specific procedures for how they determined the assay yields. Did they run the ¹H NMR analysis with the CH₂Br₂ standard before or after HPLC purification? If the latter, then this is insufficient as HPLC purification will remove any evidence of other products in the reaction. Analysis of the crude reaction mixture is absolutely required to provide a clearer picture of the reaction.

Response:

For Table 1, we had included only one example (**3aa**) of the ¹H NMR spectrum of the crude reaction mixture employed to determine the assay yields. According to the reviewer's suggestion, we have added the analysis of the crude reaction mixture and operating procedure of Table 1 in the revised Supplementary Information (Page S6-S7).

We clarified that after simple workup we directly ran the ^1H NMR analysis with the CH_2Br_2 standard. As shown in the Supplementary Figure 1 below, assay yields of **3aa** was directly calculated via the allyl proton peaks (H^1 , H^2 or H^3) by ^1H NMR spectrum of the crud reaction mixtures using CH_2Br_2 as an internal standard.

Supplementary Figure 1: ^1H NMR spectrum of the crude reaction mixtures in Table 1 (Entry 15).

For Tables 2, 3 and 4, all of the yields were the isolated yields and not the assay yields. We have labeled "Yield of isolated product after chromatographic purification" in Note [b]. We have also pointed out that all products were obtained by HPLC in the Supplementary Information.

2. The substrate scopes presented in Tables 2 and 3 are incomplete. There is only one example of an "electron-withdrawing" substituent on the 2-azaallyl framework in either table and that is the meta- CF_3 group. The para-fluoride is not truly electron-withdrawing (the authors deftly describe it as electronegative), because the Hammett resonance coefficient for a para-F is slightly negative. The meta- CF_3 group, while definitely electron-withdrawing, is not in resonance with the delocalized 2-azaallyl anion, so its impact is expected to be negligible. The authors must include at least one example of a "true" electron-withdrawing group on the para position. Additionally, it would be helpful if that group was also a competitive electrophile, such as a nitrile or ester, to explore the full substrate scope. If those substrates don't work, this needs to be reported. Another substrate that should be included in Tables 2 and 3 is a heteroaromatic ring in which the anion could delocalize onto the heteroatom (e.g. 2-

pyridyl or 2-thiazolyl). The authors do include a 3-pyridyl, which is good, but this does not really challenge the system.

Response:

According to the reviewer's suggestion, we have studied the reactions of the following imines (ketimines or aldimines) bearing electron-withdrawing and electronegative groups (4-C₆H₄-CN, 4-C₆H₄-COOMe, 4-C₆H₄-Cl, 4-C₆H₄-Br, and 3-C₆H₄-CF₃) and heteroaromatic ring (2-pyridyl, 4-pyridyl, 2-thiazolyl and 2-thiophenyl) as follows.

Screening of ketimines

Screening of aldimines

The imines (ketimines or aldimines) bearing 4-C₆H₄-Cl (**1g** or **1g'**), 4-C₆H₄-Br (**1h** or **1h'**), 3-C₆H₄-CF₃ (**1i'**) and heteroaromatic ring (2-thiophenyl, **1n**) delivered products **3ga** (45% yield, allylation of **1g**; and 43% yield, allylation of **1g'**), **3ha** (35% yield, allylation of **1h**; and 31% yield, allylation of **1h'**), **3ia** (32% yield, allylation of **1i'**) and **3na** (32% yield, allylation of **1n**). We have added these examples in Tables 2 and 3.

The imines bearing nitrile, ester, 2-pyridyl, 4-pyridyl and 2-thiazolyl, however, led to no reaction, despite increasing the reaction temperature to 60 °C or 110 °C. We added

the discussion of results in the revised paper as "With imines bearing 4-C₆H₄-CN, 4-C₆H₄-COOMe, 2-pyridyl, 4-pyridyl or 2-thiazolyl were employed, no reaction occurred." in Page 7.

3. *The solvent screening studies are also incomplete. Several studies, including those involving 2-azaallyl anions/radicals, indicate that DMSO and DMF are superior (albeit not innocent) solvents for electron-transfer processes. Results from these two solvents should be included in Table 1.*

Response:

According to reviewer's suggestion, we have tested the solvent screening studies of DMSO and DMF in the optimization table. Unfortunately, both of them did not work under the reaction conditions. We have included the results in Table 1 (entries 6 and 7).

4. *The "how" of the reaction mechanism is underexplored, particularly the role of the phenol/phenoxide leaving group. In Figure 5 and the corresponding text, the authors note that 2-naphthol is isolated in 90% yield when allyl 2-naphthyl ether (5a) is used in the reaction. This is an encouraging result that should be explored further to gain a better mechanistic insight. For example, a small collection of para-substituted phenols should be employed as substrates for the reaction and the corresponding reaction rates monitored to construct a Hammett plot analysis. The Hammett plots should follow different constants (anion resonance constant or radical constant) based on the substituents whether the phenolic group leaves as an anion (phenoxide) or radical.*

Response:

According to the reviewer's suggestion, we have studied the following *para*-substituted allyl phenyl ether such as 4-Me, 4-Ph, 4-F, 4-Br, 4-CN, 4-NO₂ for the reaction and the corresponding reaction rates monitored to construct a Hammett plot analysis. The results are as follows:

a. Competition experiments

b. Hammett plot of experimental $\log(k_{rel})$ vs. calculated $0.33\sigma + 0.67\sigma^*$

Figure 6. The Hammett plot.

We have added the discussion of results in the revised paper as "To further probe the mechanism of the allylation reaction, a Hammett study was performed using intermolecular competition experiments (Figure 6a). At the outset of these experiments, we were aware that the correlation of the relative rates might be impacted by concurrent reaction mechanisms (S_N2 , S_N2' and radical). In the event, the Hammett plots show loose correlation with typical polar substituent constant parameters ($R^2 = 0.55$ for σ , $R^2 = 0.53$ for σ^-). The fit was improved by employing the σ^* parameter ($R^2 = 0.62$) (see Supplementary Information for details). The experimental data could be better fitted to a two-parameter Hammett relationship.¹⁰⁴⁻¹⁰⁹ For example, plotting $\log(k_{rel})$ versus a combination of σ (33%) and σ^* (67%) provided a better fit ($R^2 = 0.70$) (Figure 6b), which reflects the character of the selectivity-determining and rate-limiting step that might be expected from a combination of radical character with polar influences. The ρ value determined ($\rho = +1.4$), with this combination of σ scales, is smaller than would be expected for a SET mechanism, though still consistent with the buildup of negative charge.¹⁰⁵" in Page 14.

(Ref. 104: *Adv. Synth. Catal.* **358**, 765–773 (2016); Ref. 105: *Chem. Eur. J.* **21**, 5946–5953 (2015); Ref. 106: *J. Am. Chem. Soc.* **135**, 1338–1348 (2013); Ref. 107: *J. Am. Chem. Soc.* **139**, 3876–3888 (2017); Ref. 108: *ACS Catal.* **8**, 9183–9206 (2018);

Ref. 109: *J. Am. Chem. Soc.* **105**, 1221–1227 (1983))

We have also added the construction of the above Hammett plots in the revised Supplementary Information (S24-S28).

5. The results with allyl ethers **2e** and **2j** are interesting but underexplored. In both cases, alkylation occurs at the less substituted carbon of the allylic framework to provide homoallylic imine **3ae**. But it is unclear if **3ae** is formed from **2e** by an SN2' reaction mechanism or if something else is at play. Judicious use of a ¹³C- or ²H-labelled **2e** and **2j** would provide useful insight on the likelihood/predominance of SN2' and SN2 reaction mechanisms for the respective allylations to form the same product.

Response:

According to the reviewer's suggestion, we have studied the deuterium labeling experiments (²H-labelled) as below. The allylation of *N*-benzyl-1,1-diphenylmethanimine **1a** with deuterium labeling [**D**]-**2j** and [**D**]-**2f** under the standard conditions led to the formation of [**D**]-**3ae** and [**D**]-**3af** in 71% and 46% yields. These results were consistent with those in Table 4.

OTHER ISSUES TO ADDRESS:

6. In the opening sentence, the authors quote the "escaping from the flatlands" concept in medicinal chemistry, but forget to cite the two papers from Lovering that codified that concept. These papers must be cited: F. Lovering, J. Bikker, and C. Humblet, *J. Med. Chem.* 2009, 52(21), 6752-6756 & F. Lovering, *MedChemComm*, 2013, 4, 515-519.

Response:

According to reviewer's suggestion, we have cited the above mentioned references (Ref. 1: Lovering, F., Bikker, J. & Humblet, C. Escape from flatland: increasing saturation as an approach to improving clinical success. *J. Med. Chem.* **52**, 6752–6756 (2009). Ref. 2: Lovering, F. Escape from flatland 2: complexity and promiscuity. *Med. Chem. Commun.* **4**, 515–519 (2013).) in the revised manuscript.

7. At the beginning of the second paragraph, the authors state that "homoallylic amines are found in a vast number of biologically active molecules," but it is not clear how accurate this statement is. Indeed, homoallylic amines are incredibly useful precursors for the synthesis of a vast number of biologically active molecules. But, to the best of my knowledge, there are very few examples (if any) of biologically active molecules that are just homoallylic amines similar to the one synthesized at gram-scale in this article (4aa).

Response:

According to the reviewer's suggestion, we have revised this sentence as "Homoallylic amines are incredibly useful precursors for the synthesis of a vast number of biologically active molecules." in the revised manuscript (Page 2).

8. At the end of the second paragraph, the authors forget to mention that Ni catalysis has also been used to synthesize homoallylic amines (C. Liu et al. *J. Org. Chem.* 2019, 84, 10102-10110). This paper was cited in the Supporting Information, but it is appropriate to mention it here, as well.

Response:

According to the reviewer's suggestion, we have added the mention of Ni catalysis as "various transition-metal catalysts, including Ni, Pd, Cu, Zn, Ir, Rh and Yb (Figure 1a–b).", and have cited above mentioned references (Ref. 56: Liu, C. et al. *J. Org. Chem.* 84, 10102–10110 (2019)) in the revised manuscript,

We have also mentioned Ni catalysis in Figure 1a (Page 3).

9. In the introduction, there are several examples of verb tenses not agreeing, which leads to some confusion. For example on page 3: "...we *demonstrated* that 2-azaallyl anions *serve* as SEDs and *enabled* transition metal-free...".

Response:

We have checked the verb tenses carefully. We have revised the word "serve" as "served" (Page 3), and the word "are" as "were" (Page 4) in the revised manuscript.

10. At the end of the paragraph before Table 3, the authors write "the higher yield of the aldimine **1j'** over its ketimine counterpart **1j** maybe due to the increased steric hindrance about the benzylic C-H's of the ketimine." This explanation is not very convincing. Why should it be so much different for **1j'** versus **1j** if the reactive nucleophile/radical in the reaction is the exact same delocalized species? It is possible that aldimine **1j'** is less susceptible to hydrolysis than benzophenone imine **1j**, so the reagent survives the presence of adventitious water better. Is this what they mean?

Response:

The allylation reactions were carried out under strict anhydrous and anaerobic conditions. So we can eliminate the influence of hydrolysis. Usually the aldimines are more difficult to deprotonate, because the sterics of the diphenylmethyl group. But when the ketimine **1j** is used, the deprotonation is difficult because the ortho methyl of the 2-tolyl, resulting in a decreased yield with the ketimine. It is true that both aldimine

and ketimine yield the same 2-azaallyl anion, so differences in reaction are likely due to the deprotonation step.

Referee: 2

Comments:

The entitled manuscript "Transition-metal-free allylation of 2-azaallyls with allyl ethers through polar and radical mechanisms" disclosed the novel reactivity of 2-azaallyl anions with allyl ethers under transition-metal-free conditions. The present transformation exhibits excellent substrate generality with respect to both the coupling partners to prepare homoallylic amine derivatives in good to excellent yields. Of note, straightforward gram scale preparation and simple combination of base and solvent demonstrated potential synthetic utility and ease of operation of this method. Reasonable design of mechanistic studies provided the polar and radical mechanisms process. Overall the present method is a notable addition to the chemistry of allylic alkylations and certainly satisfies both novelty and quality criteria of Nat. Commun.. The supporting information contains all the necessary experimental procedures and compounds characterization.

Some suggestions and questions for improvement of the manuscript before publication are given here below:

1. In the main text, "see Supporting Information, 'Synthesis of radical clock 7a and 11a' for details", "11a" should be "11a"; "using acetonitrile:H₂O (75:25 vol./vol.) the as mobile phase", "as the mobile phase"?

Response:

We thank the reviewer for their comments!

Thanks to the reviewer for the careful work!

We have revised "7a and 11a" as "7a and 11a" (Page 15), and "the as mobile phase" as "as the mobile phase" (Page 21) in the revised manuscript.

2. Although the reaction of 1a with 2a and 5a showed no obvious differences, aromatic rings with electronic abundant or deficient substituents should be discussed.

Response:

According to the reviewer's suggestion (also in Reviewer 1's comment #4), we have studied the aromatic rings with electronic abundant or deficient substituents (*para*-substituted allyl phenyl ether such as 4-Me, 4-Ph, 4-F, 4-Br, 4-CN, 4-NO₂) for the reaction and the corresponding reaction rates monitored to construct a Hammett plot analysis. The resulting values were plotted as logarithmic value of the conversion ratio against the corresponding σ^- and σ^+ value. The results were added in the revised paper (Figure 6).

We have also added the discussion of results in the revised paper as "To further probe the mechanism of the allylation reaction, a Hammett study was performed using intermolecular competition experiments (Figure 6a). At the outset of these experiments, we were aware that the correlation of the relative rates might be impacted by concurrent

reaction mechanisms (S_N2 , S_N2' and radical). In the event, the Hammett plots show loose correlation with typical polar substituent constant parameter ($R^2 = 0.55$ for σ , $R^2 = 0.53$ for σ^-). The fit was improved by employing the σ^+ parameter ($R^2 = 0.62$) (see Supplementary Information for details). The experimental data could be better fitted to a two-parameter Hammett relationship.¹⁰⁴⁻¹⁰⁹ For example, plotting $\log(k_{rel})$ versus a combination of σ (33%) and σ^+ (67%) provided a better fit ($R^2 = 0.70$) (Figure 6b), which reflects the character of the selectivity-determining and rate-limiting step that might be expected from a combination of radical character with polar influences. The ρ value determined ($\rho = +1.4$), with this combination of σ scales, is smaller than would be expected for a SET mechanism, though still consistent with the buildup of negative charge.¹⁰⁵" in Page 14.

We have also added the construction of the above Hammett plots in the revised Supplementary Information (S24-S28).

3. How to determine the double bond geometric configurations of **3ae**, **3af** and other similar compounds?

Response:

We have carried out the 1H - 1H NOESY experiments of the products containing carbon-carbon double bond to determine the double bond geometric configurations. The 1H - 1H NOESY spectrum of **3ae** is as follows. We found that there was no NOE effect between H^a and H^b . So, the double bond geometric configuration of **3ae** was determine as the *E*-configuration, and the structure of **3ae** was (*E*)-1,1-Diphenyl-*N*-(1-phenylpent-3-en-1-yl)methanimine. We have added the 1H - 1H NOESY spectrum and analysis of **3ae** in the revised Supplementary Information (Page S80).

All the ^1H - ^1H NOESY spectrum and analysis of other similar compounds have been added in the revised SI (Pages S71, S80, S82, S86, S88, S90, S92, S94 and S96).

In addition, the double bond geometric configuration of **3ah** was determined by the coupling constant ($J_{ab} = 15.6$ Hz). The double bond geometric configuration was determined as *E*-configuration, and the structure of **3ah** was (*E*)-*N*-(5-Methyl-1-phenylhex-3-en-1-yl)-1,1-diphenylmethanimine.

4. Light free control experiment may be needed to exclude the influence of ambient light source, such CFL promoted SET process via electron donor acceptor complexes between the coupling partners.

Response:

According to reviewer's suggestion, we have studied the reaction suggested by the reviewer as below:

In the reaction above, the allylation product **3aa** (84%) was obtained when the reaction was in the dark. This result is basically consistent with the standard reaction conditions (**3aa**, 86%). This indicates that the ambient light source has no effect on the allylation reaction, and its influence can be excluded.

Referee: 3

Comments:

*In this manuscript, Zhang and co-workers described a transition-metal-free allylation of 2-azaallyls with allyl ethers electrophiles. Although this reaction gave homoallylic amine derivatives with generally good yields, there are still some features that slightly reduce the importance of this contribution. First of all, its novelty is not high enough. The C-H bond adjacent to the imine nitrogen can undergo deprotonation to form 2-azaallyl anions with strong bases, and subsequent allylation with electrophiles is a common work, similar approach has already been reported for decades. The unique radical mechanism suggested in this manuscript only applies to some specific substrates. In fact, since the selectivity was quite good, no double bond migration isomer products were obtained (especially for **2g-2j**), two electron SN2 process is more likely. Moreover, the manuscript was not well organized. It will be better if more discussion were added, but not simply replicate the content of tables, especially in*

reaction optimization and substrate scope investigation section. In addition, the functional group tolerance was limited with respect to 2-azaallyls and high temperature was required for expansion of allyl phenyl ethers.

Overall, the referee believes the current manuscript is not suitable for Nature Communication.

Response:

Allylation of nucleophiles with highly reactive electrophiles like allyl halides can be conducted without metal catalysts. Less reactive electrophiles, such as allyl esters and carbonates, usually require a transition metal catalyst to facilitate the allylation. Allylic alkylation of carbanions can be used to prepare homoallylic amines with medicinal and synthetic value. In this article, we report a unique transition metal-free allylation strategy with allyl ether electrophiles. Reaction of a host of allyl ethers with 2-azaallyl anions delivers valuable homoallylic amine derivatives (up to 92%), which are significant in the pharmaceutical industry. Notably, the simple combination of base and solvent enabled the metal-free allylation to proceed efficiently, in which no deprotonative isomerization or cyclization of products was detected. A gram scale telescoped homoallylic amine preparation was carried out, demonstrating the potential synthetic utility of this chemistry. In addition, mechanistic studies provide insight into these C(sp³)-C(sp³) bond-forming reactions and support substrate dependent radical and anionic pathways. Unlike past advances, this allylation approach enables the synthesis of a diverse array of homoallylic amines without the addition of transition-metal catalysts, photocatalysts or organometallic reagents. These attributes increase the attractiveness of this method for applications.

In the revised manuscript, according to the reviewer's helpful suggestions, we have performed some experiments and revisions. The manuscript has been improved and the chemistry explored in more detail.

1) We have added the substrate scope of the 2-azaallyls bearing electron-withdrawing groups and heteroaromatic groups. The imines (ketimines or aldimines) bearing 4-Ph-Cl (**1g** or **1g'**), 4-Ph-Br (**1h** or **1h'**), Ph-CF₃ (**1i'**) and heteroaromatic ring (2-thiophenyl, **1n**) delivered products **3ga** (45% yield, allylation of **1g**; and 43% yield, allylation of **1g'**), **3ha** (35% yield, allylation of **1h**; and 31% yield, allylation of **1h'**), **3ia** (32% yield, allylation of **1i'**) and **3na** (32% yield, allylation of **1n**). We have added these examples in Tables 2 and 3.

2) To delve deeper into the phenolic leaving groups of the allylation reaction, a Hammett study was performed using intermolecular competition experiments. According to the reviewers' suggestions, we have studied the aromatic rings with electronic abundant or deficient substituents (para-substituted allyl phenyl ether such as 4-Me, 4-Ph, 4-F, 4-Br, 4-CN, 4-NO₂) for the reaction and the corresponding reaction rates monitored to construct a Hammett plot analysis. The results of the Hammett plot were added in the revised paper (Figure 6).

3) To further probe the radical mechanism, we have also carried out EPR experiments. Carbon centered radical intermediates were captured using spin trapping agent PBN and detected with EPR spectroscopy. The cationic signal of radical can be detected in the reaction mixtures by High Resolution Mass Spectrometry.

4) According to reviewer's suggestion, we have added more discussion of the content of tables.

We hope that the revised manuscript is now suitable for Nature Communications.

More comments are listed below:

1. The authors should consider using radical scavengers such as TEMPO to probe the radical mechanism.

Response:

According to reviewer's suggestion, we have studied the radical scavenger experiments as below. When we added radical scavengers such as TEMPO and BHT under the standard reaction conditions, TEMPO completely inhibited the allylation process, while BHT led to a reduction in reaction yield to 14%.

In addition, to further probe the radical mechanism, we have also carried out the EPR experiment. Carbon centered radical intermediates was captured using spin trapping agent PBN and detected with EPR spectroscopy as below.

b. EPR spectrum

Figure 11. X-band EPR spectrum of the PBN-trapped carbon centered radical

We have added the result in revised manuscript as "Spin trapping experiment using phenyl *N-tert*-butylnitron (PBN) as the spin trap support the proposed radical-type mechanism. Heating a mixture of **1a**, **2g**, $\text{NaN(SiMe}_3)_2$ in the presence of PBN led to formation of PBN-trapped carbon centered radical as detected with EPR spectroscopy (Figure 11). The resulting EPR signal ($g = 2.0040$, $A_N = 14.9 \text{ G}$, $A_H = 2.4 \text{ G}$) is strong and similar to other reported PBN-trapped carbon centered radicals.^{110,111} The cationic signal of radical **15** can be detected in the reaction mixtures by High Resolution Mass spectra (HRMS calc'd for $\text{C}_{16}\text{H}_{24}\text{NO}^+$ 246.1852, found 246.1851 $[\text{M}]^+$)." in Pages 19-20 and Figure 11.

(Ref. 110: *J. Am. Chem. Soc.* **138**, 8968–8975 (2016); Ref. 111: *Free Radical Bio. Med.* **3**, 259–303 (1987))

(2) For the products containing F atom, the ^{19}F NMR is normally required in the SI, the author should provide the date of ^{19}F NMR for the compounds **3fa**, **3ga**.

Response:

In the revised Supplementary Information and provided the date of the ^{19}F NMR for the compounds **3fa**, **3fa'**, **3fa''** and **3ia** on Pages S12, S13, S13 and S14.

Their ^{19}F NMR spectrum were also added on Pages S53, S55, S57 and S61 in the revised Supplementary Information.

Needless to say, we side with reviewers 1 and 2, who both viewed the manuscript and significance as novel and impactful.

Reviewers' Comments:

Reviewer #1:

Remarks to the Author:

The authors have addressed all of the concerns raised in my first review of the originally submitted manuscript. Indeed, they have gone above and beyond by conducting multiple important and decisive additional experiments. It is my opinion that, the manuscript is fit for publication in Nature Communications.

Reviewer #2:

Remarks to the Author:

The authors have addressed all the questions commented by reviewers. Complementary substrates and detailed mechanistic experiments have been added in the revised manuscript. The manuscript is recommended for publication in Nature Communications as its current version. Only one revision is suggested:

In Figure 2 (a), group "Ar" should be adjusted for its proper format.

Reviewer #3:

Remarks to the Author:

The authors have addressed all referee concerns, and is ready for publication.